# The Pharmacological Approach to Oncologic Patients with Acute Coronary Syndrome

**DOI:** 10.3390/jcm9123926

**Published:** 2020-12-03

**Authors:** Juri Radmilovic, Alessandro Di Vilio, Antonello D’Andrea, Fabio Pastore, Alberto Forni, Alfonso Desiderio, Massimo Ragni, Gaetano Quaranta, Giovanni Cimmino, Vincenzo Russo, Marino Scherillo, Paolo Golino

**Affiliations:** 1Unit of Cardiology and Intensive Coronary Care, “Umberto I” Hospital, 84014 Nocera Inferiore, Italy; juri.radmilovic@libero.it (J.R.); adivilio56@gmail.com (A.D.V.); antonellodandrea@libero.it (A.D.); f.pastore@aslsalerno.it (F.P.); a.forni@aslsalerno.it (A.F.); a.desiderio@aslsalerno.it (A.D.); m.ragni@aslsalerno.it (M.R.); g.quaranta@aslsalerno.it (G.Q.); 2Unit of Cardiology, Department of Translational Medical Sciences, University of Campania “Luigi Vanvitelli”, Monaldi Hospital, 80131 Naples, Italy; giovanni.cimmino@unicampania.it (G.C.); vincenzo.russo@unicampania.it (V.R.); 3Unit of Cardiology and Intensive Coronary Care, “San Pio” Hospital, 82100 Benevento, Italy; marino.scherillo@gmail.com

**Keywords:** cancer, acute coronary syndrome (ACS), double antiplatelet therapy (DAPT), anticoagulant, atrial fibrillation, thrombocytopenia

## Abstract

Among acute coronary syndrome (ACS) patients, 15% have concomitant cancer, especially in the first 6 months after their diagnosis, as well as in advanced metastatic stages. Lung, gastric, and pancreatic cancers are the most frequent malignancies associated with ACS. Chemotherapy and radiotherapy exert prothrombotic, vasospastic, and proinflammatory actions. The management of cancer patients with ACS is quite challenging: percutaneous revascularization is often underused, and antiplatelet and anticoagulant pharmacological therapy should be individually tailored to the thrombotic risk and to the bleeding complications. Sometimes oncological patients also show different degrees of thrombocytopenia, which further complicates the pharmacological strategies. The aim of this review is to summarize the current evidence regarding the treatment of ACS in cancer patients and to suggest the optimal management and therapy to reduce the risk of adverse coronary events after ACS in this high-risk population.

## 1. Introduction

Cardiovascular disease and cancer are responsible for the majority of deaths among persons 65 years of age and older in the Western world [1]. Amidst acute coronary syndrome (ACS) patients, 15% have concomitant cancer at various stages [2]. ACS may occur before or after establishing the diagnosis of cancer, and occasionally the diagnosis is made during hospitalization for ACS. Sometimes double antiplatelet therapy reveals the presence of a lying tumor through unexpected bleeding, such as gastrointestinal bleeding.

The incidence of ACS in patients with newly diagnosed cancer increases in the first 6 months after their diagnosis and in more advanced stages: lung, gastric, or pancreatic cancer are the most recurrent malignancies associated with ACS [3].

It is extremely important to assess the cardiovascular risk in patients with ACS and concomitant cancer: classic risk factors such as hypertension, smoking, diabetes, and obesity are somehow potentiated by several prothrombotic elements secreted by the tumor itself or expressed on its surface. Moreover, chemotherapy and radiotherapy can also exert prothrombotic, vasospastic, and proinflammatory actions [4,5]. Regarding cancer therapy, a highly prothrombotic status has been observed in patients treated with cisplatin, 5-fluorouracil, gemcitabine, erythropoietin, and bevacizumab [2].

Current guidelines for conservative and invasive treatment of ACS cannot be easily applied to all cancer patients. Indeed, prospective studies assessing the efficacy and safety of ACS treatment usually have excluded cases with active cancer. Furthermore, the management of ACS in patients with cancer is quite challenging, this is because it should be effective yet also safe. Thrombocytopenia, coagulopathy, and anemia are often present in these patients, and the probability of cancer recurrence may require interruption of double antiplatelet therapy (DAPT) in case of biopsies, surgery, or re-initiation of cancer therapy [6].

The aim of this review is to outline the current evidence in the treatment of ACS in cancer cases, as well as to hypothesize the safer management and therapy in order to reduce the risk of adverse coronary events after ACS in this high-risk population.

## 2. Methods

We performed a review of clinical studies comparing outcomes of patients with and without a history of cancer admitted for ACS and/or percutaneous coronary intervention (PCI). We achieved this by doing formal searches of the electronic database MEDLINE (source PubMed) and the Cochrane Controlled Clinical Trials Register Database. About 30 studies were identified from 2008 to 2020, by a combination of medical subject headings including the following terms: acute coronary syndrome, myocardial infarction, antiplatelet therapy, anticoagulation therapy, and cancer. References from reviews and selected articles were also examined for potentially relevant citations. Our analysis was restricted to the trials that focused on the comparison between interventional and noninterventional approach and antiplatelet/anticoagulation therapy management, with a special focus on the bleeding risk and the thrombocytopenia.

## 3. Cardiovascular Toxicities Induced by Chemotherapy and Radiotherapy

Different chemotherapeutic agents may damage the cardiovascular system, injuring coronary and peripheral circulation, with acute and long-term consequences.

5-Fluorouracil (5-FU) and its oral pro-drug Capecitabine are associated with angina and ACS: 5-FU is related with endothelial damage and alteration in molecular signaling that controls smooth muscle cell tone, inducing abnormal vasoreactivity [7]. 

Cisplatin may induce acute coronary thrombosis, determinated by endothelial damage, thromboxane production, and platelet activation and aggregation as the main mechanism [8]. In combination with cisplatin, vinblastine can induce endothelial apoptosis, while bleomicin can worsen endothelial dysfunction, increasing vasotoxic properties of these drugs [9]. 

Cyclophosphamide may induce Prinzmetal’s angina or hemorrhagic peri-myocarditis [10], while Paclitaxel and Docetaxel are associated with vasospasm, ACS, and cardiac rhythm disturbance such as bradycardia [11]. 

Vascular endothelial grow factor (VEGF) signaling pathway inhibitors may cause endothelial dysfunction, vascular remodeling, inflammation, and platelet activation, responsible for increased risk of cardiovascular events [12]. Treatment with Sinutinib is related with a reduction in coronary flow reserve and abnormal vasofunctional balance, associated with microvascular impairment and rarefication of microvascular pericytes and capillaries [13]. Sorafenib has been also associated with coronary vasospasm and progression of coronary artery disease (CAD) [14]. An increased bleeding risk has been observed in patients treated with VEGF inhibitors. 

Progression of atherosclerosis and ischemic events may be observed during treatment with Nilotinib and Polatenib, two tyrosine kinases inhibitors [15]. 

Studies [16] reported ACS and Takotsubo cardiomyopathy during Rituximab treatment. 

Moreover, increased cardiovascular risk is considered in patients with prostate cancer treated with androgen deprivation therapy (ADT), especially with gonadotropin-releasing hormone (GnRH) agonists [17,18], and in women receiving aromatase inhibitors, as anastrozole and letrozole [19]. 

Finally, radiation therapy may be related with coronary and peripheral artery disease [20]. Ionizing radiations affect noncancerous cells, and endothelial cells are the most vulnerable: within a short period after radiation exposure, cholesterol plaques and thrombosis can form, and fibrosis may involve all three layers of the vessel wall [21], see Table 1 and Figure 1. 

## 4. Clinical Presentation of ACS in Cancer Patients

The clinical presentation of ACS in cancer patients may be different from that of noncancer patients. Dyspnea is one of the most incessant symptoms (44%), followed by chest pain (30%) and hypotension (23%) [22]. Additionally, in a significant percentage of cases, ACS may begin in the form of silent ischemia, probably as a consequence of the altered perception of angina due to the neurotoxicity of some chemotherapies. Ambiguous abnormalities on electrocardiograms and increased cardiac troponin levels do not necessarily indicate ACS [2].

It was also found that about 85% of all ACS in cancer patients are non-ST segment elevation myocardial infarction (NSTEMI), whilst in women there are not infrequent cases of myocardial infarction with nonobstructive coronary arteries (MINOCA) and, in particular, Takotsubo syndrome [23].

## 5. Management of Acute Coronary Syndrome

The management of ACS in cancer patients represents a special setting in which the risk/benefit ratio should be individually evaluated. The presence of the tumor, the treatments related, and the cancer comorbidities, such as thrombocytopenia and the increased risk of bleeding, complicate the choice of the optimal treatment in the cardiovascular disease in these patients. In clinical cardiovascular trials, few data concern cancer patients, therefore the decision-making should be individualized for each patient, and the management should rely on a cardio-oncology team [24].

### 5.1. Interventional Approach

The treatment of ACS in patients with cancer constitutes a therapeutic challenge. The Society for Cardiovascular Angiography (SCAI) Expert Consensus [25] emphasized that the decision-making regarding revascularization must consider the overall prognosis of the patient: for cancer patients with an expected survival <1 year, percutaneous revascularization may be considered for patients with acute ST-elevation myocardial infarction (STEMI) and very high risk NSTEMI. For cancer patients with an acceptable prognosis, general revascularization criteria may be applied more extensively, being watchful for thrombocytopenia, vascular access complications, and excessive bleeding while undergoing DAPT [26]. For patients with stable angina, every effort should be made to optimize medical therapy before approaching with an invasive strategy; fractional flow reserve (FFR) is recommended before non-urgent percutaneous coronary intervention (PCI) to justify the need for revascularization [25].

Current guidelines do not state that invasive revascularization is not indicated in cancer [27]. However, guideline-recommended management is periodically underused: PCI is performed in only 25% of cancer cases with STEMI and in 10% of cancer patients with NSTEMI [28].

In the recent European Society of Cardiology guidelines on NSTEMI [22], the authors consider that invasive strategy could be withheld in a subgroup with nonobstructive CAD and comorbidities such as cancer.

Regarding this issue, medical literature is somewhat conflicting as various results were reported in different clinical studies.

Literature reviews [24,29] revealed that cancer patients with ACS are at higher risk of in-hospital all-cause death, cardiac death, and bleedings compared with those without cancer, as well as higher one-year all-cause death and cardiac demise. The increased risk of mortality may be explained by the numerous comorbidities, such as the prothrombotic state, the risk of stent thrombosis, and the suboptimal guideline-recommended medications received. A less recurrent use of PCI or drug-eluting stents (DES) in patients with a history of cancer admitted for ACS has been reported. The higher comorbidities associated with cancer, such as renal impairment, anemia, and bleeding risk, may contribute to the suboptimal use of invasive strategies and potent antithrombotic treatments. A tailored approach appears essential to reduce both the risk of cardiac death and bleeding during the acute phase.

Moreover, the higher rates of long-term all-cause, but not cardiac, mortality in cancer patients admitted for PCI highlights the fact that non cardiovascular comorbidities may be of greater prognostic importance over the years after an ACS, as cancer cases will mostly die of cancer in the long term [29].

A large ten-year observational study in patients with metastatic cancer and ACS suggests that PCI does not provide an additional benefit to medical therapy. In this study, Guddati et al. [28] announced that in the overall population of 49,515 patients with the metastatic disease ACS, the mortality rate was higher in the subgroup treated with PCI than in the subgroup treated with optimal medical therapy (OMT) alone; this was particularly true in NSTEMI patients. The authors therefore suggest that the decision to perform PCI in this particular clinical setting should be carefully evaluated based on individual characteristics, like the extent of other comorbidities and tumor burden.

Generally, all revascularization options are available with platelet counts more than 50,000/µL, and PCI with drug-eluting stent (DES) or bare metal stent (BMS) with a platelet count more than 30,000/µL [25].

Regarding this issue, Iliescu et al. [25] state the following:Balloon angioplasty should be advised for patients with platelet count <30,000/µL, who are not candidates for DAPT, or when a noncardiac procedure or surgery is necessary as soon as possible;BMS should be considered for patients with platelet count >30,000/µL, who need a noncardiac procedure, or surgery or chemotherapy, that can be delayed for >4 weeks;New-generation DES should be recommended for patients with platelet count >30,000/µL, who do not have an immediate need for a noncardiac procedure or surgery or chemotherapy.

However, recent studies [28,30,31] advocate that PCI + optimal medical therapy (OMT), especially acetylsalicylic acid use if not contraindicated, is associated with a better survival rate regardless of the cancer stage and/or eligibility for cancer treatment.

Cancer by itself has been proclaimed to have a higher incidence of subacute stent thrombosis: increased prothrombotic status, slower re-endothelization, and the unpredicted probability for premature DAPT discontinuation influence subacute events. The BleeMACS sub-study [32] showed that the presence of cancer in patients undergoing PCI negatively affected the prognosis and was the strongest predictor of death or reinfarction and bleedings after ACS in the one-year follow-up. Assessment of coronary stenosis with FFR is necessary before proceeding to PCI, as well as confirmation of excellent stent apposition after PCI with intravascular ultrasound (IVUS) or optical coherence tomography (OCT) [33]. OCT is recommended after stent placement to ensure optimal expansion, and an absence of complications may allow early DAPT interruption [25,26,27,28,29,30,31,32,33,34].

Plain old balloon angioplasty (POBA) is more often used in oncological patients, especially in those with a higher bleeding risk: it allows shorter DAPT time than PCI with DES [35].

Coronary artery bypass grafting (CABG) should be scheduled for 2 to 6 weeks after ACS in clinically stable sufferers with cancer and it should be performed in patients with a platelet count exceeding 50,000/μL. CABG is usually contraindicated in advanced stages due to the metastatic dissemination risk during extracorporeal circulation [36].

Transradial approach is preferable over the transfemoral one because it is safer and minimizes bleeding events, whereas the transfemoral access has a higher risk of retroperitoneal bleeding, especially if DAPT is used in a patient with thrombocytopenia. The transfemoral approach is preferable only if the transradial access had been used several times before, or if there is an abnormal Allen test, such as in women after total mastectomy and in patients on hemodialysis [37].

Finally, as suggest by Bisceglia et al. [24], in cancer patients without active bleeding, after a multidisciplinary evaluation with the oncologist related to prognosis and comorbidities, all revascularization options can be applied, with PCI appearing to be more tolerated, especially when using the newer stent platforms. Nevertheless, ACS is an acute event, therefore in acute unstable cases, a multidisciplinary evaluation of cancer prognosis may not be feasible and a cardiological invasive approach may be necessary without delay.

### 5.2. Thrombocytopenia in Cancer Patients

Thrombocytopenia, defined as a platelet count of less than 100,000/µL, is present in 10–25% of cancer patients. It can be caused by bone marrow infiltration, platelet sequestration in the spleen, increased peripheral destruction in case of sepsis, disseminated intravascular coagulation, hemolytic uremic syndrome with thrombotic thrombocytopenic purpura, heparin therapy, chemotherapy side effects, fibrinolytic drugs, Clopidogrel, and GP IIb/IIIa receptor inhibitors [38]. It should be emphasized that thrombocytopenia and thrombophilia often coexist in many clinical syndromes, despite the underlying pathophysiological mechanisms being largely unknown; it may be a consequence of compensatory increased platelet size and activation in patients with a low platelet count. Another possible cause of this paradox could be found in a major release of reticulated platelets, more active than mature ones, that may precipitate a thrombophilic state [39]. In Sarkiss et al.’s study [39], 70 cancer patients with ACS were selected, 27 of them (39%) were thrombopenic. The seven-day survival rate in patients treated with acetylsalicylic acid was higher and there were no severe bleeding complications.

The management of the dosage of antiplatelet and anticoagulant therapy in cancer patients undergoing cardiac catheterization should be adjusted based on platelet values [25]:PCIs are safe in patients with a platelet count between 40 k and 50 k/µL and no thrombotic disorders [40].The starting dose of unfractioned heparin (UFH) if the platelet count is 50,000/μL or lower should be 30 to 50 U/kg. If the platelet count exceeds 50,000/µL, UFH at a dose of 50 to 70 U/kg or bivalirudin IV should be used. If the activated clotting time (ACT) is less than 250 s during the infusion of UFH, the heparin dose should be increased. ACT monitoring is crucial for patient safety in this clinical setting [25].For a platelet count <30,000/µL, revascularization and DAPT should be decided after a multidisciplinary evaluation and a risk/benefit analysis.If the platelet count is higher than 10,000/µL, ASA can be continued; DAPT with Clopidogrel is allowed if the platelet count exceeds 30,000/µl. Prasugrel, Ticagrelor, and IIB-IIIA inhibitors should not be used if the platelet count is <50,000/µL [41].If the platelet count is <50,000/µL, the duration of DAPT may be reduced to 2 weeks after PTCA alone, 4 weeks after BMS, and 6 months after second- or third-generation DES.Prophylactic platelet transfusion can be performed if the platelet count is below 20,000/µL and the patient has a high fever, leukocytosis, a sudden decrease in the platelet count, or other coagulation disorders, or if the patient is on chemotherapy due to bladder, ovarian, colon cancer, or melanoma. Therapeutic platelet transfusion is recommended in thrombocytopenic patients who develop bleeding during or after catheterization; after, platelet transfusion is recommended to repeat platelet count [42].If the platelet count is <10,000/µL, platelet transfusion should be performed, see Figure 2.

## 6. Medical Therapy

### 6.1. Antiplatelet Therapy

Antiplatelet therapy represents a big challenge in cancer patients with ACS. These cases have a high risk of stent thrombosis and bleeding complications, which are often worsened by the presence of thrombocytopenia.

Different pathways are implicated in platelet activation in these patients, such as an increased platelet expression of adhesion molecules and direct activation by contact with molecules on tumor cell membrane [43].

In the Rohrmann et al. study [44], there was an analysis of more than 35,000 acute myocardial infarction (AMI) patients, in which it was observed that cancer patients were often undertreated (in particular with antiplatelet therapy, beta-blockers, ACEIs), which only rose in-hospital mortality.

In a retrospective study [45], a total of 456 cancer patients with an ACS diagnosis were recruited (386 of them in the form of NSTEMI). The authors suggest that approaches to the treatment of ACS in patients with thrombocytopenia should be better directed toward the evaluation of platelet function rather than toward absolute platelet count. This is because patients predisposed to coronary thrombosis may have platelets larger and more adhesive to the vascular surface.

Nevertheless, the majority of clinical studies consider the platelet count rather than their function for the management of antiplatelet therapy [31,33,44].

Thus, based on the evidence described above, the first-choice antiplatelet regimen in ACS patients with cancer should be acetylsalicylic acid and Clopidogrel.

Clopidogrel may have lower antiplatelet activity in case of liver injury possibly caused by cancer or chemotherapeutics. Ticagrelor, Prasugrel, and IIB-IIIA inhibitors should be used with more caution in cancer cases and should be avoided in patients with a platelet count <50,000/mL [31,41,42,43,44,45].

In the management of antiplatelet treatment, cancer patients with ACS may be separated into three groups, see Figure 3.

#### 6.1.1. Patients with Platelet Count >30 k/μL and Who Do not Require Urgent Surgery/Chemotherapy (in the Next 4 Weeks)

In these patients, PCI with third-generation DES is recommended, while ASA (300/75 mg) and Clopidogrel (300–600/75 mg) should be used for at least 1 month or up to 3–6 months depending on the risk of recurrent ischemia (lesion extension/procedure performed) and/or bleedings, then Clopidogrel should be discontinued and ASA maintained. In case of ASA intolerance or bleeding, it may be switched to Clopidogrel [31,46].

#### 6.1.2. Patients with a Platelet Count of 10–30 k/μL or Who have Surgery/Chemotherapy is Scheduled within the Next 4 Weeks

Balloon angioplasty is recommended, and therapy with ASA and Clopidogrel should be used for at least 2 weeks, providing there is a platelet count of >30 k/μL [24].

If urgent surgery is required, Clopidogrel should be stopped 5 days before the surgery and restarted 24–48 h after the surgery. It is recommended that ASA should not be discontinued. In case of high thrombotic risk, a bridging therapy with GPIIa/IIIb inhibitors iv started 3 days before surgery and discontinued 4–6 h before surgery may be considered [31].

In patients on DAPT and needing urgent life-saving surgery, they should receive surgery despite the excessive perioperative bleeding risk. Platelet concentrate needs to be used if required [47].

In patients who need vital chemotherapy and radiotherapy, the duration of DAPT is still debated, because data on the effect of these treatments on re-endothelization are scarce. Optimal stent extension should be considered in these patients, using IVUS or OCT [34].

#### 6.1.3. Patients with a Platelet Count of <10 k/μL

Prophylactic platelet transfusion is recommended [25,31]. In this context, there must be a careful evaluation of long-term monotherapy treatment with ASA, paying particular attention to bleeding complications and pursuing careful monitoring of platelet counts [25].

### 6.2. Triple Therapy

#### Atrial Fibrillation and ACS in Cancer Patients

One of the most complex management strategies is the treatment of the patients with cancer, atrial fibrillation, and ACS.

Cancer is present in about 2.5% of patients with nonvalvular atrial fibrillation (NVAF), and this arrhythmia is associated with a poor prognosis and in-hospital mortality. Moreover, cancer and its treatments contribute to an increased prevalence of NVAF: anticancer treatment and cancer-associated factors, such as inflammation and dehydration, augment the risk of thromboembolic complications [48].

There is little evidence regarding the use of direct-acting oral anticoagulants (DOACs) in patients with cancer and atrial fibrillation. Anticancer drugs used in clinical practice may potentiate or decrease the anticoagulant effect. For example, vinblastine, doxorubicin, sunitinib, and enzalutamide may decrease the anticoagulant effect whilst increasing the risk of embolism. Conversely, crizotinib, abiraterone, and cyclosporine tend to increase the anticoagulant effect and consequently the risk of bleeding [49].

Furthermore, in addition to the type of anticancer therapy, it is essential to pay attention to the tumor site. As previously described, the use of edoxaban and rivaroxaban is associated with an increased risk of gastrointestinal bleeding [50].

Patients with cancer who are already on an oral anticoagulant, vitamin-K antagonist (VKA), or DOACs for NVAF and who develop ACS should receive low-molecular-weight heparin (LMWH) for the first six weeks. This should be followed by, if the bleeding risk (HASBLED score <3) is low, VKA or DOAC; if the risk of bleeding is high (HASBLED score ≥3) or the prognosis is poor, it is advisable to continue with LMWH, while monitoring the blood count and renal function [51]. The anticoagulant therapy needs to be associated with the antiplatelet therapy, preferably with ASA and Clopidogrel. According to recent NSTEMI ESC Guidelines [22], after considering the ischemic and bleeding risk, the triple therapy should be maintained for 1 week to 1 month depending on the patient risk profile. After that, one of the two antiplatelet drugs is suspended (preferably ASA); from the sixth/twelfth month it is possible to continue only with anticoagulant therapy [22,49,50]. During triple therapy in cancer patients, VKA should be titrated to obtain international normalized ratio (INR) of less than 2.5 or DOAC at reduced doses [31], see Figure 4.

## 7. Complications Management: Bleedings

The occurrence of bleeding or a high risk of bleeding at baseline in patients with ACS raises hospital mortality, and the clinical consequences depend on the type and entity of bleeding.

The choice of whether to continue, temporarily discontinue, or withdraw the antiplatelet and/or anticoagulant therapy has to be based on the severity of bleeding, careful assessment of risk/benefit ratio, and the clinical scenario. In every case, it is crucial to identify the site of bleeding [49].

In case of minor bleeding (i.e., subconjunctival or mucocutaneous bleeding, hematomas), DAPT can be maintained; if the patient is on an anticoagulant, it may be continued or stopped for 24 h [50].

In the case of clinically relevant nonmajor bleeding (hemoptysis, upper or lower gastrointestinal bleeding, urinary or genital tract bleeding requiring medical approach), which does not require hospitalization, DAPT can be pursued or its duration can be reduced. In this case, oral anticoagulation should be stopped for 24 to 48 h and restarted preferably with a VKA with INR 2–2.5 or with NAO. Triple antithrombotic therapy can be replaced by dual therapy with Clopidogrel plus oral anticoagulant (OAC) [51,52,53].

In patients with severe bleeding (identified as a need for hospitalization and a decrease in hemoglobin levels >2 g/dL), DAPT must be stopped, a monotherapy with Clopidogrel should be considered and it may be reinitiated when deemed safe. Shortening the DAPT duration needs to be considered [54]. In patients on oral anticoagulation, its suspension for 5 to 7 days or the use of an antidote should be considered. In patients with mechanical heart valves, LMWH or UFH should be used for 4 to 8 days. Triple therapy has to be switched to UFH/LMWH treatment with Clopidogrel. Then, Clopidogrel alone may be used in selected cases. A PPI should be used if upper gastrointestinal bleeding is identified [31].

In case of very severe bleeding (decrease in hemoglobin levels >5 g/dL), hemodynamically stable, we should stop DAPT and continue with Clopidogrel; if bleeding persists, stopping all antithrombotic treatments must be considered. Once bleeding has ceased, the need for DAPT or single antiplatelet therapy (SAPT) may be re-evaluated, preferably with Clopidogrel. OAC treatment should be stopped and reversed until bleeding is controlled; it may be re-initiated within one week if clinically indicated. Triple therapy should be downgraded to DAPT; if the patient is on dual therapy, antiplatelet medication may be stopped if deemed safe [53,55,56,57,58].

In case of life-threatening bleeding, all antiplatelet medication should be discontinued; OAC should be stopped and reversed. Once bleeding has ceased, the need for antithrombotic treatment should be re-evaluated [50,55].

Severe bleeding requires aggressive treatment with red blood cell transfusion (Hb < 7 g/dL), considering the necessity of endoscopic or surgical procedures.

Risk-to-benefit ratio should be assessed before restarting antiplatelet and anticoagulation therapy [52], see Figure 5.

## 8. Conclusions

Cancer represents a limiting factor in the management of ACS. Every oncological case, at each stage of diagnosis and treatment, who develops ACS should follow guidelines for noncancer patients, including an invasive strategy when possible, especially in the early cancer stage and in young patients. Antiplatelet and anticoagulant therapy should be managed considering the risk-to-benefit ratio and the balance between thrombotic and bleeding risk. Moreover, chemotherapy and radiotherapy represent prothrombotic factors that may increase the thrombotic risk and compromise post-ACS therapy. In the last decade, the prognosis of cancer patients has improved, therefore optimal therapy for ACS may enhance life expectancy and reduce adverse coronary events.

## Figures and Tables

**Figure 1 jcm-09-03926-f001:**
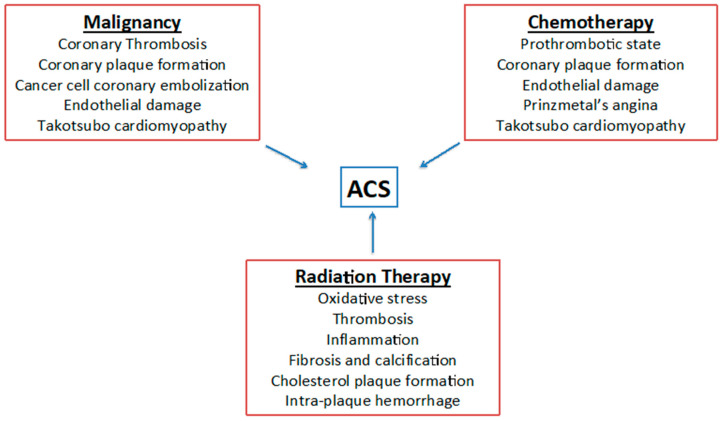
Pathogenesis of acute coronary syndrome in cancer patients. ACS: acute coronary syndrome.

**Figure 2 jcm-09-03926-f002:**
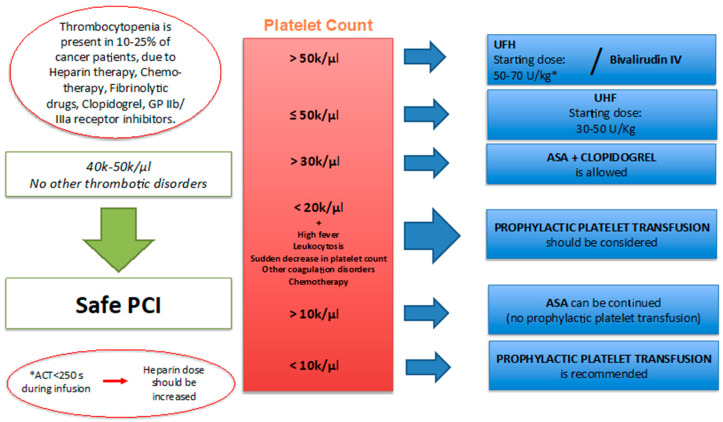
Thrombocytopenia management. PCI: percutaneous coronary intervention; ACT: activated coagulation time; UFH: unfractionated heparin; ASA: acetylsalicylic acid.

**Figure 3 jcm-09-03926-f003:**
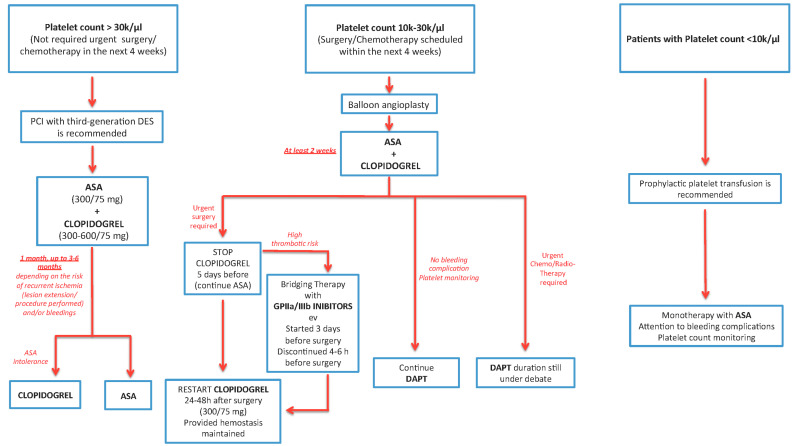
Management of antiplatelet therapy in cancer patients. DES: drug-eluting stent; ASA: acetylsalicylic acid; DAPT: dual antiplatelet therapy.

**Figure 4 jcm-09-03926-f004:**
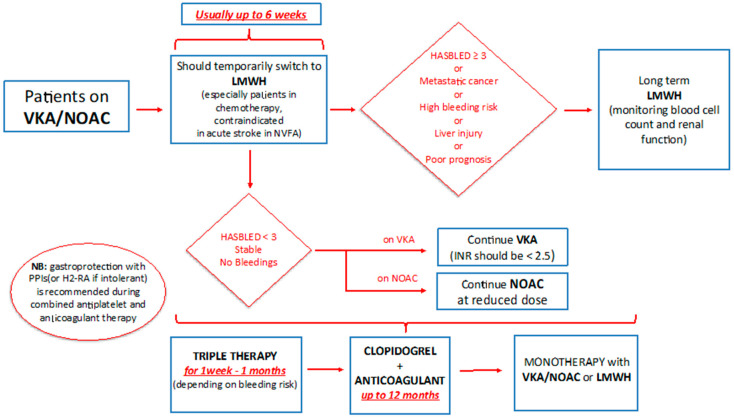
Management of anticoagulant therapy in cancer patients. VKA: vitamin K antagonist; NOAC: new oral anticoagulant; LMWH: low-molecular-weight heparin.

**Figure 5 jcm-09-03926-f005:**
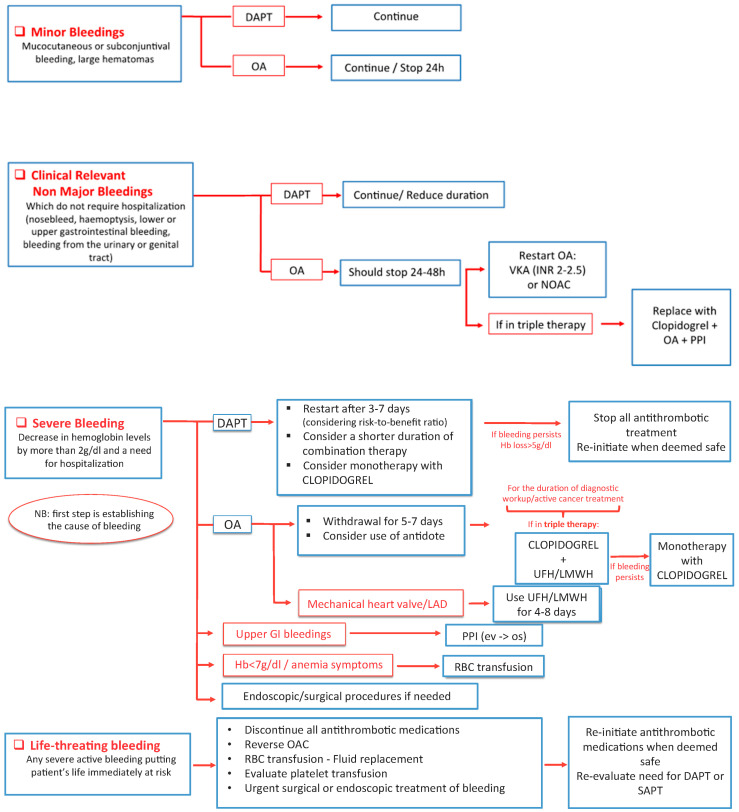
Bleeding events management. DAPT: double antiplatelet therapy; SAPT: single antiplatelet therapy; OA: oral anticoagulation; VKA: Vitamin K anticoagulant; UFH: unfractionated heparin; LMWH: low-molecular-weight heparin; NOAC: new oral anticoagulant; PPI: proton pump inhibitor; GI: gastrointestinal; RBC: red blood cell.

**Table 1 jcm-09-03926-t001:** Chemotherapeutic agents associated with acute coronary syndrome. ACS: acute coronary syndrome; CAD: coronary artery disease.

	Incidence	Presentations	Cancer Therapy
**Antimetabolites**			
5-Fluorouracil [7]	0.1–18%	Angina, Vasospasm, ACS	Colorectal, pancreas, gastric breast
**Alkylating agents**			
Cisplatin [8]	0.2–12%	Angina, coronary thrombosis, ACS, progression of CAD	Ovarian, Testicular, Bladder, squamous cell of head and neck, mesothelioma
**Antitumor antibiotics**			
Bleomycine [9]	<2%	Angina, vasospasm, ACS	Testicular, cervix, Hodgkin’s and non-Hodgkins lymphoma
**Anti-microtubule agents**			
Vinblastine [9]	<5%	Angina, ACS	Testicular, lymphoma, breast cancer
**Monoclonal antibodies**			
Rituximab [16]	<1%	Vasospasm, ACS, Takotsubo cardiomyopathy	Non-Hodgkin’s lymphoma, ChronicLymphocytic Leukemia
**Tyrosine kinase inhibitors**			
Sorafenib [14]	1%	Angina, ACS	Liver, renal cell, thyroid cancer
Sunitinib [12,13]	5–8%	ACS, Takotsubo cardiomyopathy, progression of CAD	Pancreas, renal cell, gastrointestinal stromal tumor
Nilotinib [15]	8–12%	ACS, progression of CAD, peripheral artery disease	Chronic myeloid Leukemia
**Hormone therapy**			
Aromatase inhibitors (anastrozole) [19]	2%	Angina, ACS	Breast cancer
Anti-androgens (bicalbutide) [17]	1–33%	ACS, progression of CAD	Prostate cancer
Gonadotropin-releasing hormone agonist (goserelin) [18]	1–2%	Angina, ACS	Prostate cancer
Gonadothropin-releasing hormone antagonist (degarelix) [18]	<1%	ACS	Prostate cancer

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
