# Peer review of "The Pharmacological Approach to Oncologic Patients with Acute Coronary Syndrome"

_jcm, 2020, doi:10.3390/jcm9123926_

Round 1
Reviewer 1 Report
The Authors have satisfactory addressed all concerns.
Author Response
Dear reviewer we are grateful that all concerns have been addressed.
Reviewer 2 Report
I want to thank the authors for providing a revised version of their manuscript. Which has further improved.
You redesigned the figures, and they show some improvements, but I still find most of them quite hard to follow. You utilized up to 5 font-sizes/modifications in the same figure.
e.g.: In flowchart 2 you visualize 2 parallel decision pathways - it would be logic to illustrate those as parallel decision trees from top to bottom, with the same items and the same font size and color for the same kind of item.
However this is only a suggestion, and if you wish to proceed with the current figures this is fine with me.
Author Response
Please see attachment below

This manuscript is a resubmission of an earlier submission. The following is a list of the peer review reports and author responses from that submission.
Round 1
Reviewer 1 Report
In the present review paper, Radmilovic et al provide a summary of the current evidence regarding the treatment of ACS in cancer patients. Based on the literature analyzed, authors try to suggest the optimal management and therapy to reduce the risk of adverse coronary events after ACS in this high-risk population.
This is a nice and well written article providing a schematic view of the management of antithrombotic therapy in this special population. The management of ACS in cancer patients represents a unique setting in which the risk/benefit ratio of invasive treatment should be carefully evaluated.
However, in the main purpose of this article, authors failed to include and critically analyze some recent publications such as 10.2459/JCM.0000000000000993, 10.1186/s12872-020-01352-0, 10.1097/MD.0000000000018972. Thus, authors are invited to better evaluated the current literature to differentiate their analysis from the existing ones
Author Response
Response to Reviewer 1 Comments
Point 1: In the present review paper, Radmilovic et al provide a summary of the current evidence regarding the treatment of ACS in cancer patients. Based on the literature analyzed, authors try to suggest the optimal management and therapy to reduce the risk of adverse coronary events after ACS in this high-risk population.
This is a nice and well written article providing a schematic view of the management of antithrombotic therapy in this special population. The management of ACS in cancer patients represents a unique setting in which the risk/benefit ratio of invasive treatment should be carefully evaluated.
However, in the main purpose of this article, authors failed to include and critically analyze some recent publications such as 10.2459/JCM.0000000000000993, 10.1186/s12872-020-01352-0, 10.1097/MD.0000000000018972. Thus, authors are invited to better evaluated the current literature to differentiate their analysis from the existing ones
Response 1: dear reviewer, we critically analysed and include in the text the recent publications suggested (line 339-343), differentiating their analysis from existing one in the text (line 87-100). The last publication suggested was already included in the text (reference 13).
The whole article has been edited by a person speaking native fluent English to improve the quality of the language.
We attached the revised manuscript. Please see the attachment.
Thank you
Juri Radmilovic

Reviewer 2 Report
The manuscript “The Pharmacological Approach to Oncologic Patients 2 with Acute Coronary Syndrome” by Radmilovic et al. reviews current evidence on pharmacological treatment of patients with ACS and concomitant cancer. It covers a highly actual topic. However, some aspects should be considered:
Major:
- Please include a methods section, how and when was the literature research performed, how many articles were screened?
- The information is in part redundant. I would suggest to reorganize the Manuscript in
- Clinical presentation/Indication/Interventional strategy (include Thrombocytepenia here)
- Medical Therapy (Antiplatelet Therapy, Triple-Therapy if needed)
- Complication management (Bleedings)
- I do not understand the inclusion of sections on VTE and AF in the manuscript. Of course these might occur as an additional comorbidity and complicate further management of ACS patients with cancer, but it should be included in point 2 (medical therapy). If you insist on keeping the structure and the parts I would suggest to change the title (something like cardiovascular disease in cancer patients) but then you would have to change the direction of the manuscript quite a bit.
- Antiplatelet therapy:
Although you line up the very limited evidence there is for antiplatelet therapy in cancer patients undergoing PCI, you come up with relatively concrete advises for the management. Please explain how these management strategies were developed.
- You state that one should evaluate platelet function rather than platelet count – however, you do not include platelet function in Flowchart 1. How and when should this be included in the management.
- What do you suggest for patients with a need for PCI and Platelets below 20000. You suggest to give platelets, however due to the short half-life of the transfused platelets the longerterm treatment is unclear to the reviewer? Monotherapy with ASA?
- Anticoagulation therapy:
The evidence for DOACs is increasing and you covered the most important studies. Your algorithm is rather complex. Please explain how the single factors are validated. Please compare your results with Ay et al. (https://pubmed.ncbi.nlm.nih.gov/30918939/)
- Triple therapy:
Even for patients without cancer, recent NSTEMI guidelines (which you should reference) state that triple therapy should be as short as possible. To recommend 1-6 months of triple therapy is not backed by current guidelines, especially in high-risk populations for bleeding events like cancer patients. Again, you draw conclusions from very limited data, without stating these limitations.
- Bleeding management:
With modern drug eluting stents, the risk of stent thrombosis is much lower compared to the stents used during your cited studies (publication date 2013-2014). You should probably include evaluation if DAPT is really necessary after a bleeding event.
- The figures, at least in my version are of low quality and are in part without a clear structure. Please review.
Minor:
- The first paragraph is unfortunate:
Cancer is estimated to occur in about 15% of patients with acute coronary syndrome (ACS), especially in the first 6 months from diagnosis, as well as in advanced metastatic stages. Lung, gastric and pancreatic cancers are the most frequently malignancies associated with ACS.
It is probably: Among ACS patients 15% have concomitant cancer.
Same issue with line 32: Cancer is prevalent in 15% of ACS patients.
- Please correct the spelling of months in all figures
- 150: If an urgent life‑saving surgery is required in patients on DAPT, the risk of excessive perioperative bleeding should be accepted, and platelet concentrate should be used if required.
Please revise: e.g.: In patients on DAPT and need for urgent life saving surgery, they should receive surgery despite the excessive perioperative bleeding risk. Platelet concentrate should be used if required.
Author Response
Response to Reviewer 2 Comments
The manuscript “The Pharmacological Approach to Oncologic Patients 2 with Acute Coronary Syndrome” by Radmilovic et al. reviews current evidence on pharmacological treatment of patients with ACS and concomitant cancer. It covers a highly actual topic. However, some aspects should be considered:
Major
Point 1: Please include a methods section, how and when was the literature research performed, how many articles were screened?
Response 1: dear reviewer we introduced a methods section, to explain how and when the literature research was performed and how many articles were screened (line 59-69).
Point 2: The information is in part redundant. I would suggest to reorganize the Manuscript in
- Clinical presentation/Indication/Interventional strategy (include Thrombocytepenia here)
- Medical Therapy (Antiplatelet Therapy, Triple-Therapy if needed)
- Complication management (Bleedings)
I do not understand the inclusion of sections on VTE and AF in the manuscript. Of course these might occur as an additional comorbidity and complicate further management of ACS patients with cancer, but it should be included in point 2 (medical therapy). If you insist on keeping the structure and the parts I would suggest to change the title (something like cardiovascular disease in cancer patients) but then you would have to change the direction of the manuscript quite a bit.
Response 2: dear reviewer we reorganized the manuscript, following your suggestions. We excluded the section on VTE, while we modified the section about the AF, to include it into the section about medical therapy, especially triple therapy.
Point 3:
Antiplatelet therapy:
Although you line up the very limited evidence there is for antiplatelet therapy in cancer patients undergoing PCI, you come up with relatively concrete advises for the management. Please explain how these management strategies were developed.
You state that one should evaluate platelet function rather than platelet count – however, you do not include platelet function in Flowchart 1. How and when should this be included in the management.
What do you suggest for patients with a need for PCI and Platelets below 20000. You suggest to give platelets, however due to the short half-life of the transfused platelets the longerterm treatment is unclear to the reviewer? Monotherapy with ASA?
Response 3:
Dear reviewer our management strategies proposed were based on the literature revised.
Platelet function evaluation for the management of ACS patients with cancer is considered only by Syed WY et al. in the note [24]. Nevertheless, the majority of clinical studies consider the platelet count rather than their function and for this reason we did not include the platelet function evaluation in the flowchart considered. We highlighted this point in the manuscript (line 185-191).
If platelet count is below 10000, long-term treatment consists of monotherapy with ASA, paying particular attention to bleeding complications and pursuing careful monitoring of platelet counts. We added this point in the text and in the relative flowchart (line 226-228).
Point 4:
Anticoagulation therapy:
The evidence for DOACs is increasing and you covered the most important studies. Your algorithm is rather complex. Please explain how the single factors are validated. Please compare your results with Ay et al. (https://pubmed.ncbi.nlm.nih.gov/30918939/)
Response 4: dear reviewer we modified the algorithm related, to make it less complex. Regarding the results suggested by Ay et al., they are relative to the treatment of cancer associated to venous thromboembolism, but, as you suggested before, we removed this section to keep our attention especially on the management of patients with ACS and cancer.
Point 5:
Triple therapy:
Even for patients without cancer, recent NSTEMI guidelines (which you should reference) state that triple therapy should be as short as possible. To recommend 1-6 months of triple therapy is not backed by current guidelines, especially in high-risk populations for bleeding events like cancer patients. Again, you draw conclusions from very limited data, without stating these limitations.
Response 5: dear reviewer we modified this point (line 252-257), considering the new NSTEMI ESC guidelines. We included the 2020 NSTEMI ESC guidelines in the references [31].
Point 6:
Bleeding management:
With modern drug eluting stents, the risk of stent thrombosis is much lower compared to the stents used during your cited studies (publication date 2013-2014). You should probably include evaluation if DAPT is really necessary after a bleeding event.
Response 6: dear reviewer we modified some points about this argument (line 276-293), considering recent studies about this topic. Recent studies have been included n the references [37-38].
Point 7: The figures, at least in my version are of low quality and are in part without a clear structure. Please review.
Response 7: dear reviewer we optimized the figures:
Minor
Point 8: The first paragraph is unfortunate:
Cancer is estimated to occur in about 15% of patients with acute coronary syndrome(ACS), especially in the first 6 months from diagnosis, as well as in advanced metastatic stages. Lung, gastric and pancreatic cancers are the most frequently malignancies associated with ACS.
It is probably: Among ACS patients 15% have concomitant cancer.
Same issue with line 32: Cancer is prevalent in 15% of ACS patients.
Response 8: dear reviewer we modified this point.
Point 9: Please correct the spelling of months in all figures
Response 9: dear reviewer we corrected this point (line 16 – line 32)
Point 10: 150: If an urgent life‑saving surgery is required in patients on DAPT, the risk of excessive perioperative bleeding should be accepted, and platelet concentrate should be used if required.
Please revise: e.g.: In patients on DAPT and need for urgent life saving surgery, they should receive surgery despite the excessive perioperative bleeding risk. Platelet concentrate should be used if required.
Response 10: dear reviewer we modified this point (line 219-220).
The whole article has been edited by a person speaking native fluent English to improve the quality of the language
We attached the revised manuscript. Please see the attachment.
Thank you
Juri Radmilovic

Round 2
Reviewer 2 Report
The revised version of the manuscript entitled "The pharmacological approach to oncologic patients with acute coronary syndrome" by Radmilovic et al. has improved greatly.
Especially, the logic of the whole manuscript is now adequate.
Some points should still be considered:
I would recommend to redesign the figures. Since you want to illustrate algorithms, I would endorse to design real flow-diagrams with standardized items for every figure. Right now it is quite hard to follow the instructions.
You state you performed a systematic review and meta-analysis – I agree to the review part, but you did not perform a meta-analysis.
Please use acetylsalicylic acid instead of aspirin
Author Response
Response to Reviewer Comments
The revised version of the manuscript entitled "The pharmacological approach to oncologic patients with acute coronary syndrome" by Radmilovic et al. has improved greatly.
Especially, the logic of the whole manuscript is now adequate.
Some points should still be considered
Point 1: I would recommend to redesign the figures. Since you want to illustrate algorithms, I would endorse to design real flow-diagrams with standardized items for every figure. Right now it is quite hard to follow the instructions.
Response 1: dear reviewer I redesign the figures. I realize real-flow diagrams with standardized items for every figure.
Point 2: You state you performed a systematic review and meta-analysis – I agree to the review part, but you did not perform a meta-analysis.
Response 2: dear reviewer I removed meta-analysis (line 60).
Point 3: Please use acetylsalicylic acid instead of aspirin
Response 3: dear reviewer I substituted acetylsalicylic acid instead of aspirine (line 109 - 144- 179 - 196).
Please see the manuscript in the attached file
Thank you
Juri Radmilovic
